# Giant magneto-birefringence effect and tuneable colouration of 2D crystal suspensions

Baofu Ding [1,5], Wenjun Kuang [2,5], Yikun Pan[1], I. V. Grigorieva [2], A. K. Geim [1,2 ✉], Bilu Liu [1 ✉] & Hui-Ming Cheng [1,3,4 ✉]

One of the long-sought-after goals in light manipulation is tuning of transmitted interference colours. Previous approaches toward this goal include material chirality, strain and electric-field controls. Alternatively, colour control by magnetic field offers contactless, non-invasive and energy-free advantages but has remained elusive due to feeble magneto-birefringence in conventional transparent media. Here we demonstrate an anomalously large magneto-birefringence effect in transparent suspensions of magnetic two-dimensional crystals, which arises from a combination of a large Cotton-Mouton coefficient and relatively high magnetic saturation birefringence. The effect is orders of magnitude stronger than those previously demonstrated for transparent materials. The transmitted colours of the suspension can be continuously tuned over two-wavelength cycles by moderate magnetic fields below 0.8 T. The work opens a new avenue to tune transmitted colours, and can be further extended to other systems with artificially engineered magnetic birefringence.

---

[1] Tsinghua-Berkeley Shenzhen Institute and Tsinghua Shenzhen International Graduate School, Tsinghua University, Shenzhen 518055, China. [2] Department of Physics and Astronomy, University of Manchester, Manchester M13 9PL, UK. [3] Shenyang National Laboratory for Materials Science, Institute of Metal Research, Chinese Academy of Sciences, Shenyang 110016, China. [4] Advanced Technology Institute, University of Surrey, Guildford, Surrey GU2 7XH, UK. [5] These authors contributed equally: Baofu Ding, Wenjun Kuang. ✉email: geim@manchester.ac.uk; bilu.liu@sz.tsinghua.edu.cn; hmcheng@sz.tsinghua.edu.cn

Colours produced by the interference between phase-retarded components of transmitted light are so-called transmitted interference colours[1-9]. Comparing with its counterpart from chemical pigments or dyes, the interference colour features distinctive vivid, metallic, wide colour gamut appearance and photobleaching-free advantages, attracting great interests in practical applications[3,4,9-12]. Generally, when light transmits through a birefringent medium sandwiched between two crossed polarisers, the intensity of transmitted polarised light $I$ is related to the wavelength as $I = \sin^2 \frac{\delta}{2}$, where $\delta = \frac{2\pi\Delta nL}{\lambda}$ is the phase retardation between two polarised light components, $\Delta n$ the birefringence, $\lambda$ the wavelength and $L$ the length of the optical path through the medium. With a suitable combination of $\Delta n$ and $L$, the wavelength-dependent intensity can be modulated to achieve different colours. Such spectral tuning has been achieved by electric-field-controlled birefringence effect in organic liquid crystals[5-8], and recently been extended to material chirality and strain control[3,4]. To the best of our knowledge, no attempt has ever been made to use magneto-birefringence effect to achieve the spectral tuning by magnetic field control, which has the contactless, non-invasive and energy-free edge and is, therefore, more appealing in application aspects. Such a void is likely due to difficulties in achieving sufficiently large phase retardation $\delta$, as a result of either weak magneto-optical response or ultra-low saturation magneto-birefringence of conventional transparent media[13-15]. Our analysis (see below) shows that, to observe colouration in a birefringent system, $\delta \geq 3\pi$ is needed for $\lambda$ in the visible range, which is not easily satisfied even in magnetically strongly responsive systems, such as ferrofluids, where the achievable $\delta$ is generally below $\pi$ due to opacity (limited $L$) and small $\Delta n$ due to the weak shape anisotropy of suspended ferromagnetic nanoparticles[13,14,16-18]. Two-dimensional (2D) crystals, on the other hand, offer excellent prospects for magneto-birefringent colouration, thanks to their large shape anisotropy, and a favourable combination of optical and magnetic anisotropies[19-24].

Here we show that, in the case of aqueous suspensions of 2D cobalt-doped titanium oxide (CTO), their appreciable magnetic anisotropy (arising from anisotropy of Co atoms within the surrounding crystal lattice[25-28]) forces 2D crystals to rotate until their easy magnetisation axis aligns with the magnetic field $\mathbf{H}$. The collective response of such suspended crystals induces finite birefringence because of optical anisotropy of individual 2D CTO flakes (see further and Supplementary Note 1). This magneto-birefringence tunability can be described by so-called Cotton–Mouton (CM) coefficient[15] and, for our suspensions, the coefficient reaches up to $1400\,\mathrm{T^{-2}\,m^{-1}}$, three orders of magnitude larger than that of liquid crystals (cf. Supplementary Table 1). Moreover, the relatively large saturation magneto-birefringence ($\Delta n_s \approx 2 \times 10^{-4}$, two orders of magnitude larger than for the known transparent materials exhibiting magnetic response[13,14]) results in a large $\delta \approx 6\pi$ for 10 mm optical distance at $\mu_0H = 800\,\mathrm{mT}$, allowing magnetic colouration that covers more than two-wavelength cycles in the visible range.

## Results
**Sample preparation and characterisation.** Suspensions of 2D CTO were prepared by exfoliation of bulk Co-doped $TiO_x$ crystals in water using mechanical agitation. Synthesis and characterisation of the bulk crystals is described in 'Methods' section and Supplementary Fig. 1. Briefly, bulk Co-$TiO_x$ was synthesised from a mixture of $TiO_2$, CoO, $K_2CO_3$ and $Li_2CO_3$ using an annealing and ion-exchange method. 2D CTO crystals obtained after exfoliation have a lepidocrocite-type structure, where $Co^{2+}$ ions substitute some of $Ti^{4+}$ at the octahedral sites as schematically

shown in Fig. 1a. The crystals have a large aspect ratio of $>10^3$ with a lateral size of ~1.5 µm and thickness of ~1 nm (Supplementary Fig. 1c–e). Most of the results presented below have been obtained on a 2D CTO suspension with a volume concentration of 0.02 vol%, exhibiting a high transmittance over visible $\lambda$ (Supplementary Fig. 1f). Qualitatively similar results were also obtained on more dilute suspensions.

**Magneto-birefringent colouration in 2D CTO suspensions.** Our optical setup is sketched in Fig. 1a: a quartz cuvette filled with a suspension of 2D crystals was placed between two crossed polarisers in a gradient magnetic field applied in the transverse direction, increasing from bottom to top (see 'Methods' section for details of optical measurements). When backlit with white light, a vertical rainbow of colours appeared in the cuvette, with different colours corresponding to different magnetic field strengths (Fig. 1b right, Supplementary Movie 1). To quantify the field-colour correspondence, the same suspension was placed between crossed polarisers in a uniform magnetic field. For low fields $\mu_0H < 325\,\mathrm{mT}$, as the magnetic field increased from $\mu_0H = 0$, the suspension gradually became more transparent, but no apparent colour was observed (Supplementary Fig. 2). As the field increased $>325\,\mathrm{mT}$, a colour appeared and started to evolve first from yellow to orange, then purple to blue and finally to green at the highest field of $\mu_0H \approx 800\,\mathrm{mT}$ (Fig. 1c, Supplementary Movie 1).

The observed colouration cannot be the result of the Bragg diffraction by a periodic pattern of suspended 2D crystals[12], because the transmitted colour disappeared when the polarisers were removed but the field kept on (Fig. 1b, left). Instead, as we show below, colouration is the result of magnetically induced birefringence of the suspension. In the absence of magnetic field, 2D CTO crystals are randomly oriented in the water, exhibiting an isotropic optical response with zero birefringence, $\Delta n = 0$, and the suspension remains opaque. With the field applied, the crystals start to rotate and align parallel to $\mathbf{H}$ due to their magnetic anisotropy, collectively giving rise to a finite optical anisotropy (birefringence). The mechanism responsible for the magnetically induced birefringence is illustrated in Fig. 2a. Before passing through the suspension, the linearly polarised white source can be represented by three primary colours with wavelengths $\lambda_1$, $\lambda_2$ and $\lambda_3$, of equal amplitudes. As the magnetic field is applied, 2D CTO crystals rotate and align parallel to the field, resulting in a finite birefringence of the suspension, such that the two polarised light components (parallel and perpendicular to $\mathbf{H}$) experience a wavelength-dependent phase retardation $\delta = 2\pi\Delta nL/\lambda$. When passing through the analyser, constructive interference between these two components occurs for $\delta = (2N-1)\pi$, leading to a transmission maximum ($N = 1, 2, 3$ etc. is an integer). Due to the wavelength dependence of $\delta$, if the $\lambda_2$ component transmits constructively, the $\lambda_1$ and $\lambda_3$ components are off resonance and have lower transmittance, resulting in a colour dominated by $\lambda_2$.

Quantitatively, the relation between $\Delta n$ and $H$ can be derived using a microscopic model analogous to the electro-birefringence (Kerr) effect[29]:

$$\Delta n(H) = \frac{\Delta n_s}{2}\left[3L_2\left(\frac{\Delta\chi H^2}{2k_B T}\right) - 1\right], \tag{1}$$

where $L_2(x)$ is the second-order generalised Langevin function

$$L_2(x) \equiv \frac{\int_0^\pi \cos^2\vartheta \exp(x\cos^2\vartheta)\sin\vartheta\,d\vartheta}{\int_0^\pi \exp(x\cos^2\vartheta)\sin\vartheta\,d\vartheta}. \tag{2}$$

$\Delta\chi = \chi_\parallel - \chi_\perp$ the anisotropy of the magnetic susceptibility, $k_B$ the Boltzmann constant and $T$ the temperature (see Supplementary Note 1 for detail description of our model). It follows from

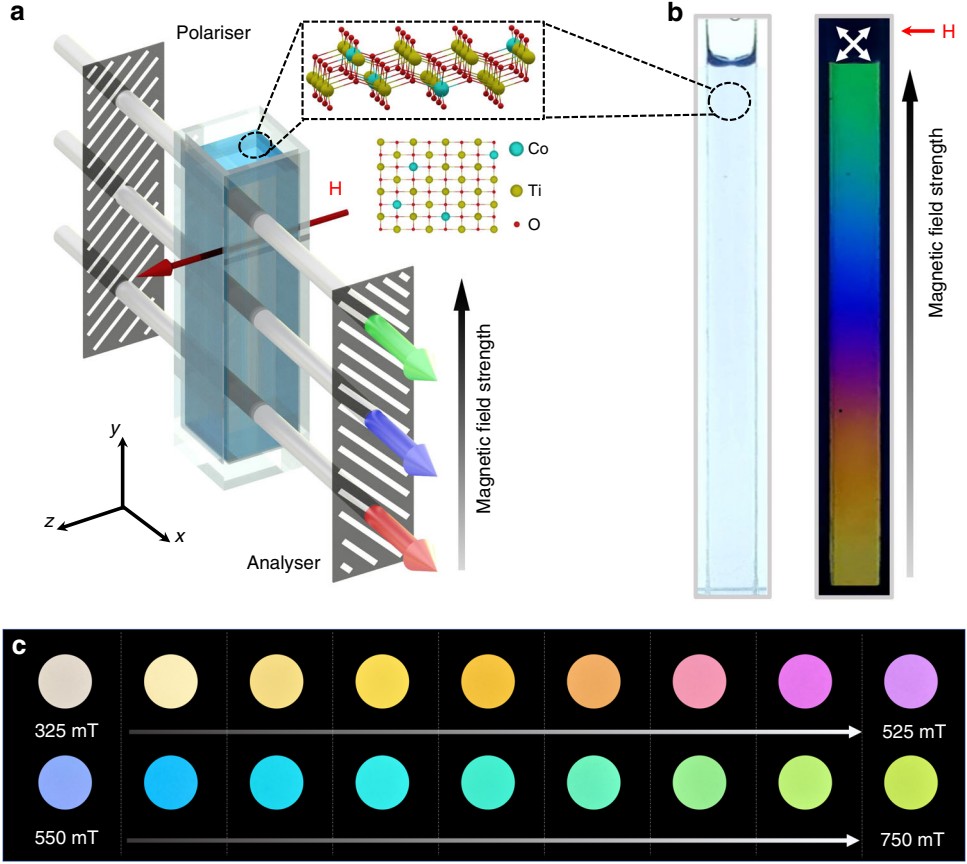

**Fig. 1 Magneto-chromatic effect in a 2D CTO aqueous suspension. a** Schematic setup of our magneto-optical experiments. A 5.5 cm tall cuvette with 1.0 × 0.5 cm rectangular cross section filled with a suspension of 2D CTO crystals is placed between two crossed polarisers, with the magnetic field applied along the $z$-direction, at 45° to the polariser/analyser. The top-right inset shows schematic atomic structure of 2D CTO. The light path is along the $x$-direction and the magnetic field is increasing vertically from bottom to top. **b** Real images of the cuvette filled with 0.02 vol% suspensions and illuminated with uniform white light. Left: no colours appear if the polariser/analyser are absent; right: a rainbow of colours is clearly seen in the presence of crossed polarisers. **c** Real images of the colours of the suspension in uniform magnetic field, as the field is tuned from $\mu_0 H = 325$ mT to 750 mT with a 25 mT step.

Eq. (1) that the birefringence increases proportionally to $H^2$ until saturation at $\Delta n_s$, corresponding to full alignment of all crystals in the suspension parallel to the magnetic field at a sufficiently high $H$. It follows that the wavelength-dependent condition for constructive interference $\delta = \frac{2\pi \Delta n(H) L}{\lambda_c} = (2N-1)\pi$ also changes with $H$, and the colour of the transmitted light is determined by $\lambda_c$, the constructive interference wavelength.

**Magnetic alignment of suspended CTO crystals**. We have verified the magnetic alignment/rotation of suspended CTO crystals by comparison of the magneto-optical transmittance when viewed along and perpendicular to the applied field (in the direction of the $z$-axis and $x$-axis in Fig. 1a, respectively). As demonstrated in Fig. 2b, the transmittance is higher in the former case, in agreement with a higher in-plane magnetisation that aligns the flakes along the field direction, so that they are parallel to the light path and allow more light through. Conversely, when the light path is perpendicular to the magnetic field, the transmission reduces to its minimum (Supplementary Fig. 3) as it is blocked by the suspended crystals, similar to window blinds. The magnetic anisotropy concluded from the optical experiments is further verified by measurements of in-plane vs out-of-plane magnetisation of our CTO crystals, showing a notably higher magnetic susceptibility for **H** parallel to their surfaces (see Supplementary Note 2, Supplementary Fig. 4). Such magnetic

anisotropy of 2D CTO is typical for paramagnetic crystals and comes from the single-ion anisotropy of Co atoms[26–28], which tend to align their magnetic moments in specific crystallographic directions. In our case, some Ti atoms are substituted by Co atoms (Fig. 1a), and their interaction with neighbouring Ti and O atoms forces Co spins to align within the TiO plane as found experimentally (Fig. 2b, Supplementary Figs. 3 and 4c).

**Tuneable spectra and CM coefficient**. For a quantitative characterisation of the observed magneto-colouration effect, we recorded transmitted spectra for different wavelengths of the incident light at different magnetic fields. The resulting map of the transmitted intensity as a function of $H$ and $\lambda$ is shown in Fig. 3a for the cross-polariser measurement geometry. As expected from the model outlined above, the transmitted intensity shows oscillations as a function of both $H$ and $\lambda$, with three transmittance maxima (white stripes in Fig. 3a) and minima (dark grey stripes) appearing alternately within the field range of 0–800 mT. Such a sinusoidal-like behaviour of the transmittance is in excellent agreement with the relation $I \propto \sin^2 \frac{\pi \Delta n(H) L}{\lambda}$, where the intensity maxima correspond to the constructive interference condition described above, whereas the intensity minima correspond to destructive interference with $\delta = \frac{2\pi \Delta n(H) L}{\lambda} = (2N-2)\pi$. Here, $N$ is the $N$th order of the intensity maximum/minimum. Based on the transmittance mapping, the corresponding

**a**

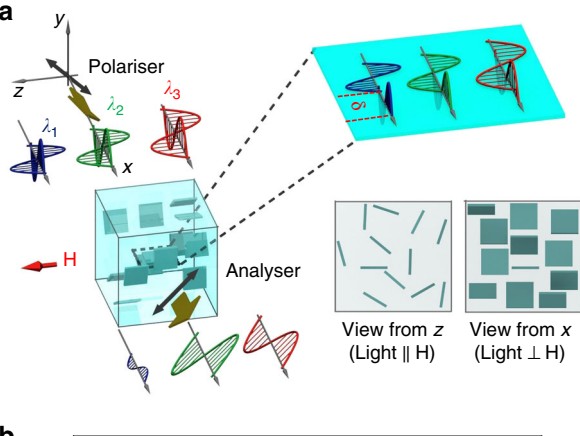

**b**

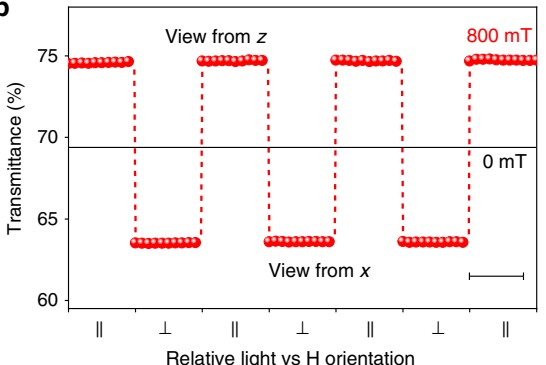

**Fig. 2 Alignment of suspended CTO crystals by magnetic field.**
**a** Schematic explanation of the colouration effect due to birefringence induced by alignment of 2D crystals (see text). **b** Transmittance of non-polarised laser light (650 nm) through the CTO suspension in the direction parallel (∥) and perpendicular (⊥) to the field of 800 mT (views from z and from x, respectively; see the bottom insets of **a**). The field direction was switched between ∥ and ⊥ once per minute. Scale bar, 50 s. The higher light transmission along the field indicates alignment of the 2D crystals parallel to it.

chromaticity diagram (Fig. 3d) shows quantitatively that the continuous colour transformation covers a wide range and forms a closed cycle within the field interval of 200–800 mT. Several individual spectra at different $H$ are plotted in Fig. 3b, where the spectral red shift induced by the magnetic field is clearly seen. For clarity, only second-order maxima are shown, corresponding to the coloured dots in Fig. 3a. It is this field-induced red shift that is responsible for the tuneable colouration reported in Fig. 1. A typical dependence of the transmitted light intensity as a function of magnetic field is shown in Fig. 4a.

Using Eq. (1) for $\Delta n(H)$, intensity $I \propto \sin^2 \frac{\pi \Delta n(H)L}{\lambda}$ and dispersion $\Delta n(\lambda)$ for the visible range (Fig. 4b), we calculated the dependence $I(\lambda, H)$ that is shown in Fig. 3c. Modelling details can be found in Supplementary Notes 1 and 3. The simulated intensity map is in good agreement with the corresponding experimental map of Fig. 3a, and the agreement is illustrated further by Fig. 4a, using a single wavelength $\lambda = 450$ nm as an example. Furthermore, we derived an intensity map from direct measurements of the phase retardation $\delta (H, \lambda)$ using spectroscopic ellipsometry, as $I = \sin^2 \frac{\delta}{2}$. This yielded an identical transmitted intensity distribution to Fig. 3a (Supplementary Fig. 5). Using the data for $\Delta n(H)$ extracted from the transmitted intensity measurements for low magnetic fields, 0–200 mT (Fig. 4c), we were able to evaluate the CM coefficient for our CTO suspensions, $C = \frac{1}{\lambda} \frac{\partial \Delta n}{\partial (H^2)}$ (ref. [15]). As is clear from the

magnetic field dependence of the phase retardation $\delta$ (Supplementary Fig. 5b), at low magnetic fields the first term in Eq. (1) is linear with $H^2$ (magnetic susceptibilities are constant at room temperature) and the CM coefficient can be found as $\approx \frac{1}{\lambda} \frac{\Delta n}{H^2}$, yielding $C$ as large as 1400 T$^{-2}$ m$^{-1}$ for $\lambda = 450$ nm, i.e., three orders of magnitude larger than, e.g., organic liquid crystals (cf. Supplementary Table 1). Such a large CM coefficient explains the particularly strong magnetic response of the CTO suspensions in our experiments, allowing tunability of the colouration by moderate magnetic fields.

## Discussion

It is instructive to discuss the conditions required to observe magneto-colouration in a birefringent system. Detailed analysis (see Supplementary Note 4) shows that, to achieve magnetic tunability of the colour of transmitted light, one needs to drive the birefringent system to at least the second-order of the constructive interference condition, which corresponds to the phase retardation $\delta \geq 3\pi$ (central white stripe in the intensity maps of Fig. 3a, c). This is because for the first-order spectral maxima (first white stripe on the left of Fig. 3a, c, where 150 mT < $\mu_0 H$ < 325 mT) the spectra of transmitted light at a single $H$ is too weakly dependent with the wavelength (Supplementary Fig. 2b) to allow discrimination of colours, and one can only detect the change of brightness of the transmitted white light, as observed in our experiments at $\mu_0 H < 325$ mT (Supplementary Fig. 2a). This also explains why in experiments with ferrofluids or magnetic liquid crystals[16–18,30], the samples became brighter (more transparent) but no colour could be observed, despite their high magnetic responsivity. Furthermore, it follows from the expression for the phase retardation $\delta = \frac{2\pi \Delta nL}{\lambda}$ that, to achieve $\delta \geq 3\pi$ at visible wavelengths, requires a sufficiently large product $\Delta nL$. This is difficult to achieve in transparent ferrofluids because of their typically small birefringence (two orders of magnitude smaller than our CTO suspensions with $\Delta n_s \sim 2 \times 10^{-4}$)[13,14] and low transparency (optical path length $L$ well <1 cm)[16–18]. We can therefore conclude that, to achieve magneto-colouration, a birefringent suspension must simultaneously meet three conditions: (1) sufficiently large magnetic anisotropy of the suspended crystals, to allow magnetic tunability of the birefringence at easily accessible fields, (2) sufficiently large birefringence $\Delta n$ and (3) high transparency to visible light (sufficiently long $L$). This analysis shows that the effect demonstrated here can be readily extended through either an increased volume fraction of the suspended crystals (Supplementary Fig. 6) or a longer optical path length (Supplementary Fig. 7), and should also be possible to achieve in other 2D magnetic suspension systems, e.g., suspensions of 2D ferromagnetic crystals[20,21,24], provided that they are sufficiently transparent to visible light.

The pronounced magneto-colouration effect exhibited by our CTO suspensions clearly demonstrates the feasibility of achieving magnetically controlled colouration using suspensions of 2D crystals as a birefringent platform. Note that magnetically controlled colouration has also been reported using light diffraction at periodic structures of suspended iron oxide nanoparticles (Bragg's law)[31]. However, the latter magneto-colouration effect has limitations and works only in the reflection mode. The resultant reflected colour is generally nonuniform and unstable because of highly sensitivity to the viewing angle, difficulties with forming stable periodic structures of nanoparticles and sedimentation induced by magnetic field. In contrast, the reported effect of magnetically tuneable transmitted colours is found to be highly reproducible and robust. The transmitted light intensity did not change over 1000 cycles of turning the magnetic field on and off (Supplementary Fig. 8). No degradation in the properties

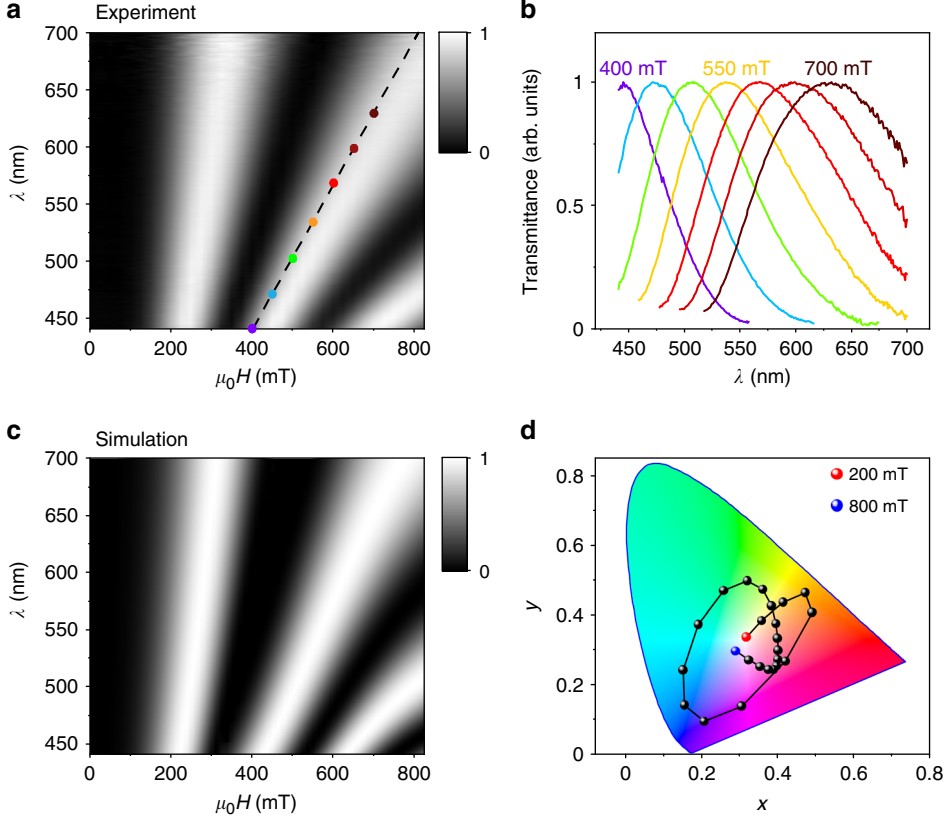

**Fig. 3 Magnetic field-dependent transmission of light through the CTO suspension at different wavelengths. a** Experimental map of the normalised transmittance as a function of $\lambda$ and $H$. The light transmission was measured using the setup with crossed polarisers as shown in Fig. 1a. Grey scale: black for zero intensity and white for the maximum intensity. **b** Red shift of the constructive interference wavelength $\lambda_c$ (corresponding to transmittance maxima) with $H$. The wavelength maxima for the colour-coded spectra correspond to the coloured dots in **a**. **c** Simulation of the transmitted light intensity based on our statistical model of magnetic-field-induced birefringence (see text). Grey scale: black for zero intensity and white for the maximum intensity. Bright and dark stripes correspond to $\delta(H, \lambda) = (2N-1)\pi$ and $(2N-2)\pi$, respectively ($N = 1, 2, 3$). **d** Coordinate evolution of transmittance spectrum in **a** during continuous increasing of magnetic field from 200 mT to 800 mT with a 25 mT step, presented at a standard CIE $xy$ 1931 colour space (CIE, International Commission on Illumination).

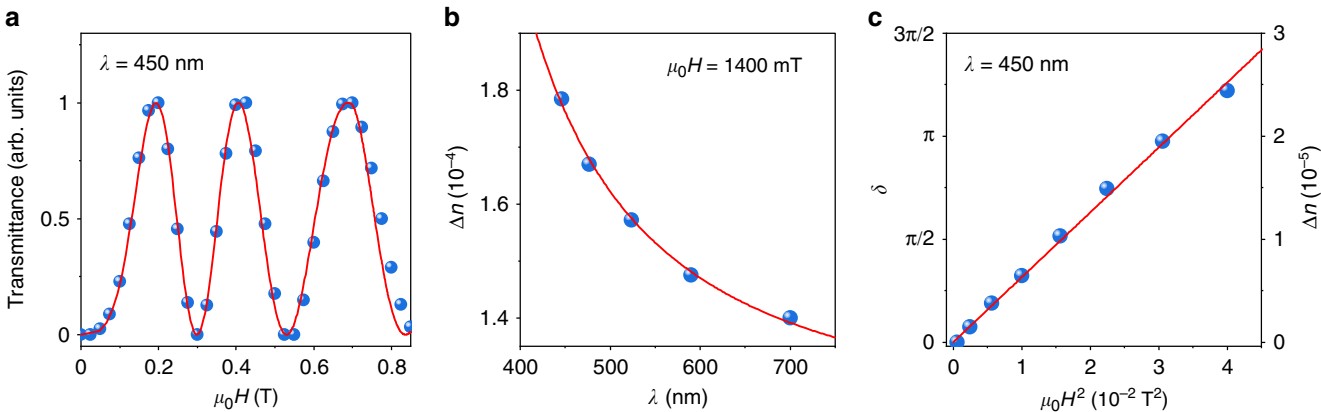

**Fig. 4 Analysis of magnetic field-induced birefringence of CTO suspensions. a** Magnetic field dependence of the light transmission for $\lambda = 450$ nm. Symbols: experimental data. The solid red curve is the best fit using the model described in Supplementary Note 1. **b** Dispersion of the birefringence. Symbols show $\Delta n(\lambda)$ calculated using $\Delta n = \lambda\delta/2\pi L$, where each value of $\lambda$ corresponds to either minimum or maximum transmission intensity measured at $\mu_0 H = 1400$ mT. Solid red line is a fit to the wavelength dependence of $\Delta n$ derived in Supplementary Notes 1 and 3. **c** Magnetic-field dependence of the phase retardation and birefringence in the low-field range, 0–200 mT. Symbols: experimental data; solid red line: linear fit. Here, $\Delta n$ is calculated from the measured $\delta(H)$ for $\lambda = 450$ nm as $\Delta n(H) = \frac{\lambda}{2\pi L}\delta(H)$.

was observed over weeks of observations, and our suspensions did not exhibit noticeable agglomeration during 2 years since the start of this work. In addition, the used long optical paths ensure enough tolerance against small variations in the optical length, which is important to achieve uniform colouration and wide viewing angles (Supplementary Fig. 8). Finally, our experiments (Supplementary Fig. 9) also showed that the colour change was fast, and the time needed to achieve a new colour did not exceed 30 ms.

In conclusion, we have observed a giant magneto-birefringence effect in transparent suspensions of 2D CTO crystals, which arises due to their large shape anisotropy, and an interplay between optical and magnetic anisotropies. The large CM coefficient together with high saturation in magneto-birefringence result in robust, uniform and reversible colouration that allows wide colour gamut controlled non-invasively by applied magnetic field. This magneto-optical effect can potentially be used in a variety of applications, including displays, magnetic field sensing, see-through printings, field-tuneable phase retarders, among many others.

## Methods

**Synthesis and characterisation of Co-TiO$_x$.** To prepare CTO monolayers, we followed a four-stage approach (Supplementary Fig. 1a), similar to previously reported but with some changes[32,33]. In stage I, TiO$_2$ (0.25 mol, 20 g), CoO (0.03 mol, 2.25 g), K$_2$CO$_3$ (0.06 mol, 5.94 g) and Li$_2$CO$_3$ (0.01 mol, 0.67 g), all from Shanghai Aladdin Biochemical Co., Ltd., China, were mixed in a stoichiometric ratio by grinding in a corundum crucible and annealed at 1000 °C for 5 h. The obtained mixture was ground again at room temperature, followed by a second annealing at 1000 °C for 20 h. The resulting compound K$_{0.8}$Ti$_{(5.2-2y)/3}$Li$_{(0.8-2y)/3}$Co$_y$O$_4$ has a layered crystal structure and good crystallinity (Supplementary Fig. 1h). In stage II, K$_{0.8}$Ti$_{(5.2-y)/3}$Li$_{(0.8-2y)/3}$Co$_y$O$_4$ (1 g) was mixed with 200 ml HCl (1 M) and agitated for 4 days, using a magnetic stirrer to allow sufficient ion exchange of Li$^+$ and K$^+$ ions with H$^+$. The protonated sediment of H$_{(3.2-2y)/3}$Ti$_{(5.2-y)/3}$Co$_y$O$_4$ (Supplementary Fig. 1b) was collected after allowing the solution to settle, washed with DI water to remove acid residues and dried in the oven (80 °C for 12 h). In stage III, protons in H$_{(3.2-2y)/3}$Ti$_{(5.2-y)/3}$Co$_y$O$_4$ were exchanged with TBA$^+$ by soaking the H$_{(3.2-2y)/3}$Ti$_{(5.2-y)/3}$Co$_y$O$_4$ powder in ~10 % (w/v) TBAOH aqueous solution (H$^+$:TBA$^+$ = 1:1 in molar ratio) for 5 h to produce TBA$_z$H$_{(3.2-2y)/3-z}$Ti$_{(5.2-y)/3}$Co$_y$O$_4$. In stage IV, the bulk compound TBA$_z$H$_{(3.2-2y)/3-z}$Ti$_{(5.2-y)/3}$Co$_y$O$_4$ was exfoliated in DI water by mechanically shaking for 48 h to obtain a stable suspension of 2D CTO.

The morphology of the intermediate products was examined using scanning electron microscopy (SEM) and atomic force microscopy (AFM; Hitachi SU8010 SEM at 5 keV, Japan and Cyper ES AFM, Oxford Instruments, UK). To analyse the elemental composition and crystallinity of the samples, we used energy-dispersive X-ray spectroscopy, high-resolution transmission electron microscopy (HRTEM, 300 kV, FEI Tecnai G2 F30, USA), powder X-ray diffraction analysis ($\lambda = 0.15418$ nm, Bruker D8 Advance, Germany) and inductively coupled plasma optical emission spectroscopy (Instrument, Arcos II MV, Germany). UV–VIS spectrophotometer (Shimadzu UV-2600, Japan) was used to determine the transmittance of the suspensions.

**Magnetisation measurements.** Magnetisation measurements were carried out using commercial Quantum Design MPMS-XL7 SQUID magnetometer. To immobilise a powder sample during the measurements, the powder was loaded in a low-magnetic-background gelatin capsule, tightly squeezed inside and sealed with Kapton tape. Preparation of the laminate of CTO crystals for measurements of magnetic anisotropy is described below. As our 2D crystals showed paramagnetic behaviour, most measurements were done at low temperatures (2 K). All magnetisation curves shown in Supplementary Fig. 4 were obtained after subtracting the background diamagnetic contribution (gelatin capsule, Kapton tape, substrate of the laminate).

The measured magnetisation $M(H)$ was analysed using standard expressions $M = N_{\text{spin}} g J \mu_{\text{B}} B_J(x)$, where $B_J(x) = \frac{2J+1}{2J} \coth\left(\frac{2J+1}{2J} x\right) - \frac{1}{2J} \coth\left(\frac{1}{2J} x\right)$ is the Brillouin function, $N_{\text{spin}}$ the total number of spins in the sample, $J$ the total angular momentum, $g$ the Landé $g$-factor ($g = 2.002$ for our paramagnetic crystals), $\mu_{\text{B}}$ Bohr magneton and $x = gJ\mu_{\text{B}}H/k_{\text{B}}T$.

**Magneto-optical measurements.** As shown schematically in Fig. 1a in the main text, a white LED or a laser with a specific wavelength was used as the incident light source. The sample (suspension of 2D CTO in 0.5 cm × 1.0 cm cross section and 5.5 cm tall quartz cuvette) was placed between two crossed polarisers (extinction ratio 10$^5$:1, Glan-Laser Calcite Polarisers, GL10-A, Thorlabs Inc.), with a magnetic

field supplied by a home-made electromagnet applied in a direction perpendicular to the optical path and at 45 degrees to the polariser/analyser. The magnetic field was measured using a Hall sensor. Transmitted light was detected by a spectro-radiometer (PR-788, Photo Research) equipped with a MS-75 lens, allowing detection at longer distances in order to avoid the effect of magnetic field on spectroradiometer.

## Data availability
The data that support our findings are available upon reasonable request from B.L.

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

## Acknowledgements

We acknowledge support by the National Natural Science Foundation of China (Nos. 51920105002, 51991343, 51722206 and 51521091), the Youth 1000-Talent Program of China, the Guangdong Innovative and Entrepreneurial Research Team Program (No. 2017ZT07C341), the National Key R&D Program (2018YFA0307200), the Bureau of Industry and Information Technology of Shenzhen for the '2017 Graphene Manufacturing Innovation Center Project' (No. 201901171523) and the Shenzhen Basic Research Project (JCYJ20190809180605522). The work was also supported by EPSRC grant Grand Challenges, Lloyd's Register Foundation and the Royal Society (UK). We acknowledge Xiaoyuan Hou, Xiaolong Zou, Shuqing Zhang, Shaohua Chen, Beibei Lu, Xingke Cai, Hao Xu, Tianshu Lan, Yuting Luo and Shuxiao Chen for help in some experiments and discussions.

## Author contributions

B.D., B.L., H.-M.C. and A.K.G. designed and directed the project. Y.P., B.D. and B.L. synthesised the materials and performed materials related characterisation and discussions. W.K., B.D., I.V.G. and A.K.G. carried out the magnetic property characterisation and analysis. B.D. performed magneto-optic experiments. W.K., A.K.G., B.D. and I.V.G. performed the theoretical modelling and simulations. B.D., W.K., B.L., I.V.G., A.K.G. and H.-M.C analysed the data and co-wrote the paper with feedbacks from other authors.

## Competing interests

The authors declare no competing interests.
