## [Peer Review File · Nature Communications]

REVIEWER COMMENTS

Reviewer #1 (Remarks to the Author):

The manuscript presents interesting results on magnetic birefringence in magnetic 2D crystal suspensions. The authors demonstrate giant effect that leads to tunable colouration based on phase retardation. The results are presented rigorously and carefully, so they can be of interest to wide community of researchers, even including graduate students. The demonstrated effect has possible applications in the areas of medicine, communication devices, sensing, etc.

However, there are some questions and comments.

1) What are arbitrary units for transmittance in Fig.2 and how are they related to those in Fig. 3 and in Supplementary figures?

2) I recommend to include the definition of the generalized Langevin function to the main text of the article, since there are different definitions in literature.

3) There are several misprints in Supplementary in Eqs. (1),(2),(3),(5) and (6): ' \geq ' instead of ' $>=$ ', ' $>$ ' instead of ' $>-$ ', 'd' instead of ' $d\theta$ '.

In general, I consider the manuscript deserves publishing.

Reviewer #2 (Remarks to the Author):

The manuscript reports a strong magnetic field tunable coloration in aqueous suspensions of 2D Co-TiOx. This coloration is the result of the birefringence of the suspension, which can be explained by the B-field tunable crystal orientation due to the magnetic anisotropy in the Co-TiOx. Such coloration covered the visible spectrum within a relatively small change in magnetic field. The authors further propose theoretical models and evaluate the Cotton-Mouton coefficient and cross compared with different systems, which is about several orders of magnitude larger than other transparent liquid. The manuscript is concluded by listing out the key requirement for the tunable coloration due to magneto-birefringence.

Overall, I find these new results interesting and the interpretations are reasonable. And the discovery may provide different aspects of utilization of 2D material and may further expand the research in this field. Therefore, I recommend for publication in nature communication after some minor issues are addressed.

Below are my specific comments:

1. Regarding the magnetic alignment mechanism. The authors use both optical transmittance and magnetic susceptibility measurement to show that the crystal orientation aligned with magnetic field. However, some of these measurements need to be further discussed to give readers a more direct correlation between the measurements and the conclusion:

a. In fig 2b, the authors present the transmittance along different axes. But only two axes are represented with $\sim 15\%$ total transmittance changes. It is recommended that the authors show that indeed the view from z is at the maximum and view from x is at the minimum of the transmittance.

b. To support their claim, the authors also measured the magnetization along different axis in the bulk crystal. From supplementary fig.3, the material seems to be a weak ferromagnet. Even though magnetization indeed shows difference between $H_{//}$ surface and H_{\perp} surface, but this may not be a strong evidence to distinguish hard/easy axis.

In order to further justify the claim, I would recommend that the author present some crystal orientation analysis with respect to different H direction (for example SAXS measurement) for give a direct and convincing evidence. If this is too difficult to conduct, the authors may need explain why it is challenging to do so, and introduce reader why magnetic field can align the crystal

structure.

2. In figure 3a and c, the authors compared the experiment data and the simulation. It would be more convincing if the author can compare directly the magnetic field dependence of the spectra. For example, overlapping the calculated result with the experiment one in fig 4a.

3. For the greyscale maps in Fig. 3a, c and supplementary Fig. 4a, Fig.5 and Fig.6 please include scale bars.

4. The author mentioned that the traditional diffraction induced coloration is usually non-uniform and unstable due to the highly sensitive to viewing angle. Is there any angle dependence of the coloration in this study?

5. It would be nice if the authors can include some cycling test and see how reproducible and recyclable the suspension is.

Reply to Reviewer #1:

The manuscript presents interesting results on magnetic birefringence in magnetic 2D crystal suspensions. The authors demonstrate giant effect that leads to tunable colouration based on phase retardation. The results are presented rigorously and carefully, so they can be of interest to wide community of researchers, even including graduate students. The demonstrated effect has possible applications in the areas of medicine, communication devices, sensing, etc.

We thank the Reviewer for the positive assessment of our work.

However, there are some questions and comments.

1) What are arbitrary units for transmittance in Fig.2 and how are they related to those in Fig. 3 and in Supplementary figures?

Thank you for pointing out this omission. The arbitrary units referred to the power of the transmitted laser light. The figure's axis and caption have now been changed accordingly. Fig. 2 and Supplementary Fig. 1f show optical transmittance of light through our magnetic suspensions with no polarizers used, whereas Fig. 3 describes more detailed measurements using crossed polarizers (see Fig. 1a) as a function of wavelength and magnetic field. Following the Reviewer's comment, we have realized that this difference was poorly described. The revised captions for all the three figures now clarify this point.

2) I recommend to include the definition of the generalized Langevin function to the main text of the article, since there are different definitions in literature.

Following the comment, we have added the definition of the Langevin function as eq. 2 in the revised main text.

3) There are several misprints in Supplementary in Eqs. (1),(2),(3),(5) and (6): '≥' instead of '>=', '>' instead of '>-', 'd' instead of 'dθ'.

We are grateful for noticing these misprints. They have been corrected.

In general, I consider the manuscript deserves publishing.

We appreciate this recommendation.

Reply to Reviewer #2:

The manuscript reports a strong magnetic field tunable coloration in aqueous suspensions of 2D Co-TiOx. This coloration is the result of the birefringence of the suspension, which can be explained by the B-field tunable crystal orientation due to the magnetic anisotropy in the Co-TiOx. Such coloration covered the visible spectrum within a relatively small change in magnetic field. The authors further propose theoretical models and evaluate the Cotton-Mouton coefficient and cross compared with different systems, which is about several orders of magnitude larger than other transparent liquid. The manuscript is concluded by listing out the key requirement for the tunable coloration due to magneto-birefringence. Overall, I find these new results interesting and the interpretations are reasonable. And the discovery may provide different aspects of utilization of 2D material and may further expand the research in this field. Therefore, I recommend for publication in nature communication after some minor issues are addressed.

We are grateful for this positive evaluation of our work.

Below are my specific comments:

1. Regarding the magnetic alignment mechanism. The authors use both optical transmittance and magnetic susceptibility measurement to show that the crystal orientation aligned with magnetic field. However, some of these measurements need to be further discussed to give readers a more direct correlation between the measurements and the conclusion:

a. In fig 2b, the authors present the transmittance along different axes. But only two axes are represented with ~15% total transmittance changes. It is recommended that the authors show that indeed the view from z is at the maximum and view from x is at the minimum of the transmittance.

We fully agree with this recommendation. Following the Reviewer's comment, we have added a new experimental figure (Supplementary Fig. 3). It reports the experiment suggested by the Reviewer. The results prove that the minimum transmission indeed occurs for the x-axis view. As described in the caption of Supplementary Fig. 3, our setup allows rotational measurements within $\pm 30^\circ$ around the x-axis. Unfortunately, similar measurements around the z-axis are not possible (see Supplementary Fig. 3) because the electromagnet has a small hole in its poles, which provides light access only in the direction parallel to the magnetic field (z-axis).

b. To support their claim, the authors also measured the magnetization along different axis in the bulk crystal. From supplementary fig.3, the material seems to be a weak ferromagnet.

Even though magnetization indeed shows difference between H//surface and H_{||} surface, but this may not be a strong evidence to distinguish hard/easy axis.

In order to further justify the claim, I would recommend that the author present some crystal orientation analysis with respect to different H direction (for example SAXS measurement) for give a direct and convincing evidence. If this is too difficult to conduct, the authors may need explain why it is challenging to do so, and introduce reader why magnetic field can align the crystal structure.

We appreciate the comment and apologize for not making this point clear.

Our material Co-TiO_x shows purely paramagnetic response. This follows from the magnetic field dependence in Supplementary Fig. 4a that is accurately described by the Brillouin dependence. Even for a very weak ferromagnet, such a fit would be impossible yielding unphysically large values of the angular momentum J (our fit yields $J = 3/2$ as expected for a paramagnetic material). Also, ferromagnets usually exhibit magnetic hysteresis. No hysteresis was observed in our experiments even at 2 K. As an additional evidence, the temperature-dependent susceptibility of our samples clearly followed the Curie-Weiss law as expected for paramagnets (see Figure R1a below).

Figure R1| **a**, Temperature-dependent susceptibility of the Co-TiO_x powder. Symbols: experimental data. The red curve represents the best linear fit by $1/\chi \propto (T + \theta)$ yielding $\theta \approx 0$ and indicating no ferromagnetic transition in this material down to liquid-helium temperatures. **b**, Macroscopic alignment in a highly concentrated Co-TiO_x suspension (10 g/L). For clarity, the green rectangles in the right panel highlight streaks inside the suspension, which show that 2D crystals aligned parallel to the magnetic field.

Despite being paramagnetic, our Co-TiO_x crystallites exhibit notable magnetic anisotropy. This often happens if atoms responsible for the magnetism tend to align their moments in a specific crystallographic direction (new refs 26-28). In our case, some Ti atoms are substituted by Co atoms (Fig. 1a of the main text). Their interaction with nearby Ti and O atoms forces Co spins to align in TiO planes as found experimentally (Fig. 2b, Supplementary Figs. 3 and 4c). In addition, this crystalline (single-ion) anisotropy was visually confirmed in the following two experiments. First, a Co-TiO_x laminate sample (2D crystallites piled on top of each other) was suspended inside our electromagnet (Supplementary Fig. 4). The field forced the laminate to rotate in such a way that 2D crystallites became parallel to the field (inset of Supplementary Fig. 4c). This proves that the easy direction for our paramagnetic Co-TiO_x crystals was in plane. Second, if we used a highly concentrated Co-TiO_x suspension (10 g/L), the same magnetic alignment could also be seen, even in suspension (Figure R1b).

As for the suggested SAXS measurements, this would require a dedicated measurement setup that we do not have. We believe that our many experiments (described above and in the manuscript) prove consistently and beyond doubt the existence of magnetic anisotropy for Co atoms within TiO lattice which forces the crystals to align in magnetic field. We hope that the Reviewer would agree with this assessment. We have clarified the discussed physics and the origin of magnetic anisotropy in the revised manuscript (first paragraph on page 2 and the bottom of page 5) and, also, added refs 26-28.

2. In figure 3a and c, the authors compared the experiment data and the simulation. It would be more convincing if the author can compare directly the magnetic field dependence of the spectra. For example, overlapping the calculated result with the experiment one in fig 4a.

We appreciate the suggestion, and the corresponding changes were made in Fig. 4, the main text on page 7 and in SI (last paragraph on page S2). To avoid repetition, old Figs. 4a and 4b

are replaced with new Fig. 4a that compares the experimental data (extracted from Fig. 3a) with the theoretical fit.

3. For the greyscale maps in Fig. 3a, c and supplementary Fig. 4a, Fig.5 and Fig.6 please include scale bars.

The scale bars have been added for all the figures. Thank you.

4. The author mentioned that the traditional diffraction induced coloration is usually non-uniform and unstable due to the highly sensitive to viewing angle. Is there any angle dependence of the coloration in this study?

The colouration exhibited by our system is retained over a wide viewing angle from 0 to 45°. This valuable suggestion to improve the manuscript has been added as new Supplementary Fig. 8a.

5. It would be nice if the authors can include some cycling test and see how reproducible and recyclable the suspension is.

Following the Reviewer's comment, we have carried out the suggested experiment (Supplementary Fig. 8b and page 10 of the main text). The suspension shows excellent cyclability. Thank you.

REVIEWERS' COMMENTS:

Reviewer #1 (Remarks to the Author):

I'm satisfied with the revision of the manuscript and consider that it deserves publication in the present form.

Reviewer #2 (Remarks to the Author):

In the updated manuscript, the authors did extra experiments to further justify their claim. The connection between magnetism and crystal orientation are explained, and the analysis and the data presentation are further refined. Overall, my question and suggestion are well addressed. Therefore, I enthusiastically recommend the publication of this manuscript in nature communications.

Response to Reviewer #1

Comment. I'm satisfied with the revision of the manuscript and consider that it deserves publication in the present form.

We thank the Reviewer for the positive assessment and recommendation of our work.

Response to Reviewer #2

In the updated manuscript, the authors did extra experiments to further justify their claim. The connection between magnetism and crystal orientation are explained, and the analysis and the data presentation are further refined. Overall, my question and suggestion are well addressed. Therefore, I enthusiastically recommend the publication of this manuscript in nature communications.

We are grateful for this positive evaluation and reviewer's recommendation of our work.